# Genome-Wide Identification and Evolution of the GRF Gene Family and Functional Characterization of *PbGRF18* in Pear

**DOI:** 10.3390/ijms241914690

**Published:** 2023-09-28

**Authors:** Rongxiang Zhu, Beibei Cao, Manyi Sun, Jun Wu, Jiaming Li

**Affiliations:** 1State Key Laboratory of Crop Genetics and Germplasm Enhancement, College of Horticulture, Nanjing Agricultural University, Nanjing 210095, China2018104035@njau.edu.cn (M.S.); 2Guangxi Key Laboratory of Plant Functional Phytochemicals and Sustainable Utilization, Guangxi Institute of Botany, Guangxi Zhuang Autonomous Region and Chinese Academy of Sciences, Guilin 541006, China; 3Zhongshan Biological Breeding Laboratory, Nanjing 210014, China; 4Academy for Advanced Interdisciplinary Studies, Nanjing Agricultural University, Nanjing 210095, China

**Keywords:** pear, GRF, gene structure, phosphorylation, gene expression, *PbGRF18*

## Abstract

Proteins encoded by the G-box regulating factor (GRF, also called 14-3-3) gene family are involved in protein–protein interactions and mediate signaling transduction, which play important roles in plant growth, development, and stress responses. However, there were no detailed investigations of the GRF gene family in pear at present. In this study, we identified 25 GRF family members in the pear genome. Based on a phylogenetic analysis, the 25 GRF genes were clustered into two groups; the ε group and the non-ε group. Analyses of the exon–intron structures and motifs showed that the gene structures were conserved within each of the ε and non-ε groups. Gene duplication analysis indicated that most of the PbGRF gene expansion that occurred in both groups was due to WGD/segmental duplication. Phosphorylation sites analysis showed that the main phosphorylation sites of PbGRF proteins were serine residues. For gene expression, five *PbGRF* genes (*PbGRF7*, *PbGRF11*, *PbGRF16*, *PbGRF21*, and *PbGRF23*) were highly expressed in fruits, and *PbGRF18* was highly expressed in all tissues. Further analysis revealed that eight *PbGRF* genes were significantly differentially expressed after treatment with different sugars; the expression of *PbGRF7*, *PbGRF8*, and *PbGRF11* significantly increased, implying the involvement of these genes in sugar signaling. In addition, subcellular localization studies showed that the tested GRF proteins localize to the plasma membrane, and transgenic analysis showed that *PbGRF18* can increase the sugar content in tomato leaves and fruit. The results of our research establish a foundation for functional determination of PbGRF proteins, and will help to promote a further understanding of the regulatory network in pear fruit development.

## 1. Introduction

Pear (*Pyrus* ssp.) is classified in the tribe Maleae of the subfamily Amygdaloideae in the botanical family Rosaceae [1]. Pears have a >3000 year cultivation history [2]. Pears originated in southwest China and spread throughout central Asia and then to western Asia, and eventually to Europe [2]. Except for parasitic plants, most plants on earth are known to fix carbon by photosynthesis and produce soluble sugars as alkaline carbohydrates by various reactions in the cytoplasm. These sugars provide energy to power cellular processes, one of which is the synthesis of diverse biopolymers, such as proteins and nucleic acids. Sugars are also involved in the regulation of various metabolic pathways and biological and abiotic stress responses in plants, where they function as signaling molecules [3]. Additionally, sugars as a quality trait can contribute to the fruit flavor, and sugar content is a quantitative trait controlled by multiple genes. Until now, only a few genes related to fruit quality were identified [4,5,6]. As a result of the rapid development of DNA sequencing techniques, more and more plant genomes have been sequenced in the past twenty years, including pear [7,8,9,10], laying a solid foundation for genome-wide analysis of *GRF* genes in pear.

The GRF proteins were first isolated from bovine brain tissue in 1967, and are now known to be ubiquitous in eukaryotes, and the genes are expressed in virtually all tissue types [11,12]. GRF proteins consist of multiple isoforms, and were named “G box factor GRF”, with “GF14” being the first in *Arabidopsis*, and the GRF isoforms in other plants were also named G-box regulating factors (GRFs) because they were parts of protein/G-box complexes [12,13]. Structural studies have shown that the GRF proteins consist of nine typical antiparallel α-helices [14]. They often form homodimers or heterodimers, and each GRF protein in the dimer is able to interact with a different protein [15]. This property allows them to bring two different proteins together as a protein complex. Thus, GRFs play important roles in several biological processes, such as protein movement, protein interactions, and protein stability [16,17,18,19].

An in vitro assay showed that GRFs can bind to metabolic enzymes and affect the sugar content in plants [20]. Examples were the tomato GRF proteins TFT1 and TFT10, which might modulate sucrose phosphate synthase (SPS) activity to regulate sugar metabolism during fruit development [21]. Indeed, the activity of SPS at the different stages of fruit development might be downregulated by GRFs [21]. Inhibition of six 14-3-3 genes in transgenic potato plants increased the activities of SPS and nitrate reductase [22]. In addition, GF14f–RNAi grains exhibited higher levels of several sugars (fructose, sucrose, and glucose) compared with WT in rice [20]. In addition, sugar starvation-regulated *MYBS2* and GRF protein interactions enhance plant growth, stress tolerance, and grain weight in rice [23].

There is an increasing body of evidence to show that GRF proteins play important roles in different aspects of plant physiology. For example, most GRF genes are expressed at high levels during fruit development and postharvest ripening in banana, indicating that they may participate in fruit development [24]. *Lr14-3-3* was found to be involved in the defense response of lily against the fungal pathogen *Fusarium oxysporum* [25]. Expression of a pear 14-3-3a gene (*Pp14-3-3a*) was found to be regulated during fruit ripening and senescence, and it was involved in the response to salicylic acid and in ethylene signaling [26]. In Arabidopsis, leaf starch accumulation increased when the expression of two genes, *AtGRF10* and *AtGRF9*, was reduced using antisense technology, indicating that GRF proteins regulate starch synthesis [27]. The transcription factor AtGRF5 and the transcription co-activator AN3 regulate cell proliferation in the leaf primordia of *Arabidopsis thaliana* [28]. Overexpression of the mango *GRF* genes, *MiGF6A* and *MiGF6B*, promotes early flowering in transgenic Arabidopsis plants [29]. GRF proteins, through interaction with CDPK (calcium-dependent protein kinase), regulate plant growth and development, including plant metabolism, hormone synthesis, flowering, and other biological pathways [18]. A 14-3-3 protein interacts with CPK21 to strongly stimulate its kinase activity, which can result in increased GORK phosphorylation and changes in activity-induced K^+^ efflux in *Arabidopsis* [30].

Due to their multiple roles in growth and development, the physiological functions of GRF proteins were of great interest in plant science. To date, genes encoding GRF family members have been identified in diverse plant species by whole-genome studies. For example, there were 13 *GRF* genes in *Arabidopsis* [12], 8 in rice [31], 25 in banana [24], 11 in grape [32], 17 in wheat [33], 9 in *Citrus sinensis* [34], 16 in mango [35], and 26 in *Camellia sinensis* [36]. Despite the fact that *GRF* genes have been characterized in other plants, little is known at present about the GRF gene family in pear species.

Here, we performed a genome-wide identification and characterization of GRF proteins in pear and other plant species. The expression patterns in tissues and in response to sugar signaling were characterized by using publicly available RNA-seq and qRT-PCR data. Our results explore the evolutionary relationships of GRF family proteins in nine plant species and provide a theoretical basis for future studies of the biological functions of GRF gene family members in pear.

## 2. Results

### 2.1. Identification of Pear GRF Genes

In the present study, a total of 25 GRF family members were identified from the pear genome, and the relevant physicochemical properties of the individual members, which were named *PbGRF1* to *PbGRF25* (Table 1), were determined. In addition, we identified 26 MdGRFs, 8 FvGRFs, 11 PmGRFs, 10 PpGRFs, 21 PcGRFs, 9 RoGRFs, 11 PavGRFs, 8 OsGRFs, and 13AtGRFs (Appendix A). Pear Chr10 contained four PbGRF genes (*PbGRF7*, *PbGRF8*, *PbGRF21*, and *PbGRF22*), while chromosomes 5, 11, 13, and 17 contained only a single *PbGRF* gene each. In addition, *PbGRF2*, *PbGRF3*, *PbGRF15*, *PbGRF20*, and *PbGRF25* were located on a single scaffold. The lengths of the deduced PbGRF proteins ranged from 55 (*PbGRF12*) to 899 (*PbGRF8*) amino acid (aa) residues, and the GRF proteins in the other eight plant species ranged from 60 aa (PavGRF9) to 748 aa (PcGRF11). The predicted MWs (molecular weights) of the PbGRFs ranged from 6.19 kDa (PbGRF12) to 99.97 kDa (PbGRF8), which was similar to the GRF proteins from other Rosaceae species (6.77 KDa–81.99 KDa). The predicted pIs (isoelectric points) of the PbGRF proteins ranged from 4.43 (PbGRF2) to 9.39 (PbGRF9), and these results showed that the pIs of most GRF proteins were <7.0, indicating that the proteins were acidic (Appendix A).

### 2.2. Phylogenetic Analysis of the GRF Proteins

To gain a deep understanding of the evolutionary histories and phylogenetic relationships of the GRF proteins from the nine species, we re-constructed a phylogenetic tree using the PbGRF, AtGRF, OsGRF, PavGRF, RoGRF, MdGRF, PmGRF, FvGRF, and PpGRF protein sequences (Figure 1). The *GRF* gene families were conserved in these nine plants, and the proteins could be classified into the ε and non-ε groups with strong bootstrap support (Figure 1A). PbGRF4, PbGRF5, PbGRF6, PbGRF8, PbGRF9, PbGRF15, and PbGRF20 were clustered on one branch, indicating that they may be derived from WGD/segmental duplications. In addition, PmGRF2, MdGRF19, and RoGRF4 were located close to this branch, showing protein conservation in the Rosaceae species. However, the PbGRF genes were found to be weakly related to the AtGRFs, and most of them were located on different branches. We found that 76% of the PbGRFs were classified into the ε group, which was much higher compared to the proteins from the other plant species (25% in rice, 38.46% in Arabidopsis) (Figure 1B). Furthermore, the PbGRF proteins were more closely related to the MdGRF proteins, and were clustered together on the same branches, suggesting that GRF genes have been more conserved during the evolution of woody plant species than in the evolution of herbaceous plant species.

### 2.3. Structural and Motif Distribution Analyses of the GRF Genes and Proteins

The above results showed the evolutionary and phylogenetic relationships among the GRF proteins from eight other plant species. Next, we analyzed the gene structure of all GRF proteins, because gene structures correlate with assignments to the ε group and the non-ε group. The exon–intron patterns, in terms of the number of introns and exon lengths, were obviously different between the two groups of GRF genes, suggesting diversification of the GRFs during evolution. To gain insights into the GRF family, an unrooted phylogenic tree was constructed for the 121 members (Figure 2A). The results showed that GRF proteins clustered into the same branch had similar motifs as well as gene structure.

Next, we further performed motif prediction for the GRF proteins and identified 10 different motifs (Motifs 1–10), ranging in length from 15 to 43 amino acids (Appendix A). Most GRF members in the same subclade had similar motifs, suggesting that they may have similar functions. The majority of GRF proteins contained six motifs (except motifs 7, 8, 9, and 10); however, PavGRF9, MdGRF17, PmGRF2, PmGRF9, PpGRF2, PbGRF12, and PbGRF25 only contained a single motif (Figure 2B). Most ε group members had motifs 1, 2, 3, 4, 5, 6, and 10, while non-ε group members lacked motif 10. This result provided further evidence to support the division of the PbGRF protein family into two clades. To identify gene structures and the evolutionary trajectories of the GRF genes in pear, we investigated the exon–intron compositions of the 121 *GRF* genes from the nine species. Most members of the non-ε group had 4 exons and 3 introns. A large diversity in exon number was found in the ε group; the number of exons in the *PbGRF* genes ranged from 1 (*PbGRF12*, *PbGRF17*, and *PbGRF25*) to 12 (*PbGRF8*), but most *GRF* genes in the other eight species had 4 to 6 exons (Figure 2C).

### 2.4. Synteny Analysis of the GRF Genes

The evolution and expansion of gene families were closely related to the occurrence of tandem and segmental duplications. Repeated episodes of small-scale and large-scale gene duplication events play important roles in the expansion of gene families. The most common segmental duplication event in plants leads to the expansion of family members on different chromosomes [37]. Our results indicate that the 25 *PbGRF* genes were distributed unevenly on eleven chromosomes and scaffolds in the pear genome (Table 1 and Figure 3A). To understand the duplication events that shaped the GRF gene family in the pear genome, we performed a collinearity analysis. The PbGRF1/PbGRF7, PbGRF10/PbGRF22, PbGRF11/PbGRF19, and PbGRF13/PbGRF23 gene pairs were generated by WGD/segmental duplications, and they were located on different chromosomes (Figure 3A). Our analysis showed that 64% of the PbGRF genes were generated by WGD/segmental duplication, which was the major driving force for *PbGRF* gene expansion, and the other *PbGRF* genes (36%) showed dispersed duplication (Appendix A). Additionally, a total of 15 pairs of orthologous genes were identified between pear and apple, which account for the largest number (Figure 3B). In contrast, rice only shared 2 orthologous gene pairs with pear; this strongly supports the close evolutionary relationship between the *PbGRF* and *MdGRF* genes, and was consistent with the fact that apple and pear evolved from a common ancestor.

### 2.5. CREs in the GRF Gene Promoter Regions

In the promotor of genes, CREs (cis-regulatory elements) can bind transcription factors which then act to regulate gene expression. Therefore, we characterized the CREs present in the promoter regions of the *GRF* genes; the DNA sequences 2000 bp upstream of the transcription initiation sites of GRF genes were defined as the putative promoter region. A total of 19 CREs were identified, including drought-inducibility, light responsive (six elements), gibberellin (GA) responsive (two elements), methyl jasmonate responsiveness (MeJA responsiveness), low-temperature responsive, salicylic acid responsive, abscisic acid (ABA) responsive, defense and stress responsiveness, auxin-responsive, metabolism regulation, anaerobic induction, and meristem expression (Table 2, Appendix A). Interestingly, we found that the CRE involved in light-responsiveness was the most prevalent element in the *GRF* gene promoters of the nine species, and we identified 1251 light-responsive elements. However, the CRE involved in abscisic acid responsiveness was only found in the promoter regions of the *AtGRF* and *FvGRF* genes (Table 2). There were 282 light responsive elements in 25 *PbGRF* gene promoters, which accounted for the largest proportion of all *PbGRF* promoters. The CRE involved in MeJA responsiveness was predicted in 18 *PbGRF* promoters (Appendix A). Our study found a diversity of CRE distribution patterns in the promoter regions of the *GRF* genes, suggesting that the expression of GRF genes may be regulated by various factors, such as, light, ABA, MeJA, and GA response elements.

### 2.6. Phosphorylation Site Analysis

Plant GRF proteins play a central role in the web of phosphorylation [38] and have been reported to be phosphorylated on serine, threonine, and tyrosine residues [39]. This phosphorylation can induce conformational changes in proteins, resulting in changes in protein activation or functional status. Whether the phosphorylation regulation of GRF proteins occurs during development in pear was not well understood at present. In this study, the phosphorylation sites of all PbGRF proteins were analyzed (Figure 4), and the results showed that the main phosphorylation site in the PbGRF proteins was serine. This result was consistent with previous studies showing that in GRF proteins, the well-described phosphorylated residues were Ser and Thr [40,41].

### 2.7. Expression Characteristics of GRF Genes in Different Tissues of Pear

As we know, GRF genes play diverse functional roles in different tissues. In order to characterize the functions of GRF genes in pear, gene expression was profiled in different tissues and in six developmental stages of pear fruits. Of all 25 *PbGRF* genes, expression of six genes was not detected in any tissue (Appendix A), and expression of nine genes was not detected at any stage of pear fruit development (Appendix A). Five *PbGRF* genes (*PbGRF7*, *PbGRF11*, *PbGRF16*, *PbGRF21*, and *PbGRF23*) were found to be highly expressed in fruits (Figure 5A), indicating that those GRFs might play important roles in fruits. However, *PbGRF18* was highly expressed in all tissues. In the middle and late developmental stages of pear fruits, we found that most *PbGRF* genes were highly expressed, except for *PbGRF8* and *PbGRF24*, which were expressed in young fruits (Figure 5B), and this expression pattern was also supported by the qRT-PCR results (Figure 6). Combining the transcriptome and qRT-PCR results, we found that *PbGRF* genes might be positively involved in fruit growth and development, and that *PbGRF8* participates in the embryonic development of pear fruits.

### 2.8. PbGRF Gene Expression Can Respond to Sugar Content

To understand the potential function of *PbGRF* genes in sugar signaling pathways, we examined their expression levels in different sugar treatments by qRT-PCR. Seedlings of the pear species ‘duli’ (*Pyrus betulifolia* Bunge) were treated separately with 2% glucose, 2% fructose, and 2% sucrose (water was the negative control) for one month. Among the *PbGRF* genes, the expression levels of *PbGRF11* in the fructose, glucose, and sucrose treatments were significantly higher than in the water control, and the expression levels of *PbGRF1*, *PbGRF7*, and *PbGRF8* were significantly increased in the fructose treatment than that of water treatment (Figure 7). The expression levels of *PbGRF3*, *PbGRF7*, and *PbGRF8* were higher in the sucrose treatment. The above results suggest that *PbGRF1*, *PbGRF3*, *PbGRF7*, *PbGRF8*, and *PbGRF11* may be involved in sugar signaling (Figure 7).

### 2.9. PbGRF18 Could Regulate Plant Growth and Fruit Sugar Accumulation

To determine the intracellular localization of the three PbGRF proteins, PbGRF8::GFP, PbGRF11::GFP, and PbGRF18::GFP fusion constructs were transiently expressed in *Nicotiana benthamiana* leaf cells. The fluorescent signals from PbGRF8 and PbGRF18 were observed in the cytoplasm and plasma membrane; however, the fluorescent signal from PbGRF11 was observed in the cytoplasm, plasma membrane, and nucleus (Figure 8, Appendix A). The subcellular localizations of the PbGRF proteins in *N. benthamiana* suggest that they may be highly active on the plasma membrane and in the cytoplasm of pear cells.

Expression analysis has shown that *PbGRF18* was highly expressed in different tissues and fruit development. In order to confirm the hypothesis that *PbGRF18* has effects on plant growth and fruit sugar accumulation, we used transformation of the tomato cultivar ‘Micro-Tom’ to increase the expression levels of *PbGRF18*. The ‘Micro-Tom’ plants expressing the OE-PbGRF18 construct grew significantly better than wild-type plants (Figure 9A,B), and the fructose and glucose contents of the fruits and leaves were also significantly increased (Figure 9C,D). In conclusion, our results strongly suggest that *PbGRF18* was involved in plant growth and play important roles in sugar accumulation.

## 3. Discussion

### 3.1. Identification and Characterization of GRF Genes in Pear

Genes encoding GRF proteins have been found in all eukaryotes examined to date, and they usually consist of multiple isoforms [11,12]. The GRF proteins interact with phosphorylated targets to act as phosphor-binding regulators in signaling transduction pathways [18,19]. Therefore, GRF proteins play important roles in developmental regulation in plants. As far as we know, the largest plant GRF gene family reported was identified in *Brassica napus* with 46 *BnaGRF* genes [42], and *Triticum aestivum* (wheat) ranked second with 30 *TaGRF* genes [43]. Pear was one of the most important fruit tree crops, but the studies focused on the *GRF* genes in pear are still very limited. Here, we identified 25 *PbGRF* genes in the pear genome, which was similar to the number of *GRF* genes present in the banana and cotton genomes [24,44].

Based on the amino acid sequences, gene structures, and phylogenetic analysis, the 25 PbGRF proteins were classified into two groups (Figure 1 and Figure 2). This result was also in accordance with previous studies in Arabidopsis and soybean [11,45]. The gene exon/intron characteristics were crucial to understanding gene function and evolutionary relationships [46]. Previous studies have shown that the proportion of intron-free genes in different eukaryotic organisms varies from 2.7% in the nematode *Caenorhabditis elegans* to 97.7% in the mammalian pathogen *Encephalitozoon cuniculi* [47]. The intron-free gene *AcAPX10* plays an important role in the stress response process in *Actinidia chinensis* (kiwifruit) [46]. In pear, the *PbGRF12* gene, in the same way as *AcAPX10*, might also play an important role in the stress response process, because it does not require intron splicing prior to transcription.

### 3.2. Evolution History of the PbGRF Gene Family

Tandem duplication, segmental duplication, and WGD/segmental duplication events were important contributors to the expansion of gene families during plant evolution. For example, the expansion of the WRKY and AP2/ERF gene families were primarily driven by WGD/segmental and tandem duplication events [48]. The SWEET gene family was thought to have expanded through WGD/segmental and dispersed duplications [49]. Our results further elucidated the mechanisms of expansion of the *GRF* gene family in pear. The results showed that 64% of the PbGRF genes were generated by WGD/segmental duplication, which was the major driving force for *PbGRF* gene expansion, and the other *PbGRF* genes (36%) showed dispersed duplication. These findings suggest that homologous genes were formed gradually during the development of plants, avoiding the event that plant growth was slowed owing to the loss of function of a single gene [50]. Additionally, the number of chromosomes in the pear genome was affected by recent whole-genome duplication events, resulting in an increase from 9 chromosomes to 17 [37,51], also resulting in the PbGRF gene members’ expansion.

### 3.3. CREs and Phosphorylation Sites Identification in GRF of Pear

CREs play an important role in the regulation of gene expression. A CRE analysis showed that *GRF* genes were involved in the light response and in abiotic stress responses in plants [34,35]. In Arabidopsis, GRF protein was specifically bound to phototropin 1 (Phot1) under blue light [52], and under red light, At14-3-3µ and At14-3-3ν interact with photoperiod regulatory proteins [53]. In this study, we found that all *PbGRF* genes contain light response elements (Table 2, Appendix A), indicating that they might participate in light regulatory pathways, and were closely related to plant growth and development.

In plants, GRF proteins display specific target binding properties [12,54,55], and regulate important biological processes through the functional modulation of a wide array of target proteins via protein–protein interactions. GRF proteins were characterized as phosphothreonine/phosphoserine binding proteins; the target binding sites were RSXp-SXP, RSXXpSXP, and pS (R is arginine, S is serine, and P is proline, X is any amino acid, and pS is phosphoserine) [16]. GRF proteins had been shown to be phosphorylated at multiple sites in a different plant species [56], indicating that phosphorylation was a probable mechanism that regulates the functionality of the GRF proteins. In Arabidopsis, a total of 17 different phosphorylation sites have been characterized in eight distinct GRF isoforms [57,58], with site-specific phosphorylation of certain GRF residues, thought to be mediated by calcium-dependent protein kinases. In the present study, phosphorylation sites were predicted in the PbGRF proteins, and the most serine sites were detected, indicating that the biological functions of PbGRF proteins were modified by posttranslation modification events (Figure 4).

### 3.4. Functional Characterication of PbGRF18

Previous studies had shown that GRF proteins were phosphorylated by calcium-dependent protein kinases (CPKs) [59,60], and we also found that *PbCPK28* could promote sugar accumulation in pear fruits, in our previous research [61]. There was a potential interaction mechanism between PbCPK28 and PbGRFs, which together regulate the accumulation of sugar content in pear fruits. In our study, we found that the expression levels of most *PbGRF* genes increased continuously during fruit development (Figure 6), which was consistent with the general trend of sugar accumulation, indicating that PbGRF proteins may participate in sugar accumulation. Among all PbGRFs, PbGRF18 was highly expressed in different pear tissues, including pear fruit, so the transgenic tomato plants were observed. Finally, our results proved that *PbGRF18* contribute to the sugar accumulation during fruit development. Additionally, PbGRF18 also promoted rapid plant growth during plant development (Figure 9). Combined with our previous research result, we speculate that the mechanism of GRF promoting fruit sugar accumulation in pear might be through phosphorylated PbCPK28, thus enhancing the PbCPK28 activity, and increasing fruit sugar accumulation, finally.

## 4. Materials and Methods

### 4.1. Plant Materials

The ‘Duli’ (*Pyrus betulifolia* Bunge) seedlings were collected from an orchard in Hebei province, China, and were grown in a growth chamber at Nanjing Agricultural University, under a 16  h/8  h light/dark photoperiod with 75% relative humidity at 25 °C for 30 days, and used for the sugar treatments. After thirty days of treatment, the leaves of the ‘Duli’ seedlings were flash-frozen and stored at −80 °C prior to analysis. Each sample had at least three biological replicates. *Nicotiana benthamiana* plants were grown in a growth chamber with 16  h/8  h light/dark at 23 °C. The *N*. *benthamiana* seedlings were grown until they had more than six leaves.

The pear cultivar ‘Dangshansuli’ (*Pyrus bretschneideri*) used for qRT-PCR was collected from an orchard in Fengxian in the Jiangsu province, China. Five different tissues (petal, ovary, anther, leaf, and stem) and five fruit stages (35, 49, 78, 90, and 140 DAFB (days after full bloom)) of ‘Dangshansuli’ were collected from the same trees. The samples were frozen in liquid nitrogen and then stored at −80 °C for further analysis.

### 4.2. Identification of GRF Genes in Pear

Pear genome data were obtained from the ‘Dangshansuli’ (*Pyrus_bretschneideri* Rehd) pear genome database [8]. The Hidden Markov Model (HMM) profiles of the GRF motifs (PF00244) were downloaded from the Pfam database (http://pfam-legacy.xfam.org, accessed on 30 April 2023 version 35.0) [62] and used to search the GRFs from the pear genome sequence using hmmer software (version 3.3.2). In order to further identify the potential GRFs in pear, protein sequences were submitted to the Pfam database for further validation, and genes encoding proteins with the GRF motifs and e-value < 1 × e^−5^ were retained for further analysis.

### 4.3. The Phylogenetic Tree Construction

The predicted sequences of GRF proteins from Arabidopsis (*Arabidopsis thaliana*), rice (*Oryza sativa*), sweet cherry (*Prunus avium*), red raspberry (*Rubus idaeus*), apple (*Malus domestica*), plum (*Prunus mume*), strawberry (*Fragaria × ananassa* Duch.), and peach (*Prunus persica*) were downloaded from Phytozome v12.1 (http://www.phytozome.org, accessed on 30 April 2023) [63]. These proteins were named: pear (PbGRF), Arabidopsis (AtGRF), rice (OsGRF), sweet cherry (PavGRF), red raspberry (RoGRF), apple (MdGRF), plum (PmGRF), strawberry (FvGRF), and peach (PpGRF). All protein sequences were aligned using ClustalW (http://www.clustal.org/clustal2/#Contact, accessed on 30 April 2023 version 2.1) [64], and the alignment was then used to construct a neighbor-joining phylogenetic tree with 1000 bootstrap replications using Mega 11.0.13 [65]. The phylogenetic tree was visualized using Evolview tools (http://evolgenius.info/#/, accessed on 30 April 2023).

### 4.4. Gene Duplication and Collinearity Analysis

The gene duplication landscape was obtained using t MCScanX [66]. The putative duplicated genes were linked by connecting lines. The Plant Genome Duplication Database (http://chibba.agtec.uga.edu/duplication/, accessed on 30 April 2023) was used to perform synteny analysis between pear and *Arabidopsis*. Moreover, according to previously published criteria, gene duplication events were defined based on their chromosomal locations: genes located on the same chromosome were considered to result from tandem duplications, and genes located on different chromosomes were called segmentally duplicated genes [37].

### 4.5. Analysis of Cis-Regulatory Elements (CREs) in the GRF Genes

To investigate the transcriptional control mechanisms of the PbGRF genes, we used Bedtools (https://bedtools.readthedocs.io/en/latest/content/installation.html accessed on 30 April 2023 version: v2.27.1) to extract the 2 kb upstream sequences (putative promoter region) and the PlantCARE (http://bioinformatics.psb.ugent.be/webtools/plantcare/html/, accessed on 30 April 2023) webtool was used to identify the CREs in the potential promoter regions. TBtools was used to visualize the identified regulatory elements.

### 4.6. Prediction of Phosphorylation Sites, Protein Motifs, and pI

MEME (Multiple Em for Motif Elicitation) (https://meme-suite. org/meme/tools/meme, accessed on 30 April 2023) was used to examine the conserved protein motifs, with the number of motifs set to 10 [67]. The relative molecular weights (MWs) and isoelectric points (pI) of the deduced PbGRF proteins were predicted using the ExPASy proteomics server database (http://expasy.org/, accessed on 30 April 2023). The NetPhos 3.1 Server (https://services.healthtech.dtu.dk/service.php?NetPhos-3.1, accessed on 30 April 2023) was used to analyze the phosphorylation sites, with a threshold of 0.5.

### 4.7. Expression Profiles of PbGRF Genes Determined from RNA-seq Datasets

Transcriptome sequencing (RNA-seq) data from six different developmental stages of ‘Dangshansuli’ pear fruits and seven different tissues of ‘Dangshansuli’ were acquired from our previous study [68]. A K-means clustering method was used to normalize the FPKM (fragments per kilobase of transcript per million mapped reads) values of the *PbGRF* genes using R 3.2.2 software [69], and the pheatmap package was used for heatmap visualization (http://mac.R-project.org/tools, accessed on 30 April 2023).

### 4.8. RNA Extraction for Gene Cloning and qRT-PCR Analysis

Total RNA was isolated using an RNA extraction kit (Vazyme Biotech Co., Ltd. Nanjing, China), followed by first-strand cDNA synthesis for qRT-PCR using *TranScript*^®^ One-Step gDNA Removal and cDNA Synthesis Supermix (TransGen Biotech Co., Ltd. Beijing, China). qRT-PCR analysis was performed using LightCycler^®^ 480 SYBR Green I Master mix (Roche) according to the manufacturer’s protocol. The pear *PbGAPDH* (*Pbr036263.1*) gene was used as the internal control for the normalization of gene expression.

### 4.9. Subcellular Location

The full-length coding sequences of *PbGRF8*, *PbGRF11*, and *PbGRF18*, without the stop codons, were cloned into the pCAMBIA1300 vector. *A. tumefaciens* strain GV3101 was transformed with the three gene vectors and then infiltrated into young leaves of *N. benthamiana*. Plants were then incubated in a growth chamber with a 16 h light/8 h dark photoperiod at 25 °C/23 °C for three days. Fluorescence signals were observed with a laser confocal microscope (Zeiss, Oberkochen, Germany). All of the experiments were performed with three biological replicates.

### 4.10. Micro-Tom Transformation and Soluble Sugar Quantification

The transformation protocol used for the tomato cultivar ‘Micro-Tom’ was that of Sun et al. [70]. Soluble sugars were extracted as described by Li et al. [61].

### 4.11. Oligonucleotide Primers

The names and nucleotide sequences of all primers used in the present study were given in Appendix A.

### 4.12. Statistical Analyses

All values obtained from the studies were presented as the mean ± SEM. Significant differences were determined by one-way ANOVA tests in Figure 6 and Figure 7. The significances differences were calculated using a Student’s *t*-test in Figure 9, which were indicated with asterisks, (****) *p* < 0.0001.

## 5. Conclusions

The results of our study provide a reference point for subsequent studies investigating the functions of *PbGRF* genes. The analysis of the structural characteristics of the GRF gene family members in nine plant species using a variety of bioinformatics software was conducive to the further study of *GRF* gene function. *PbGRF* genes have extensive expression profiles which span multiple developmental stages and sugar signaling, implying their crucial roles in various physiological functions. We found that *PbGRF18* promoted sugar accumulation and regulated plant growth and development in pear. In summary, our findings provide new clues that will be useful for improving the sugar content of pear fruits and laid the groundwork for further comprehensive analysis of PbGRF proteins.

## Figures and Tables

**Figure 1 ijms-24-14690-f001:**
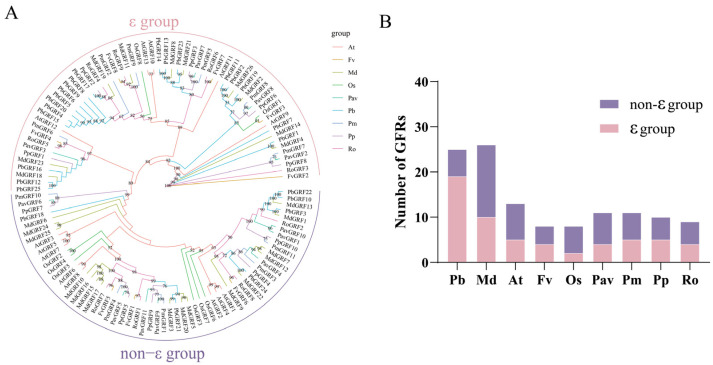
Phylogenetic analysis of GRF proteins. (**A**) The phylogenetic tree illustrates the evolutionary relationships among 121 GRF proteins from nine plant species (25 PbGRFs, 8 OsGRFs, 11 PavGRFs, 9 RoGRFs, 26 MdGRFs, 11 PmGRFs, 8 FvGRFs,10 PpGRFs, and 13 AtGRFs). The unrooted neighbor-joining tree was constructed using the LG model as implemented in MEGA 11.0.13, and it grouped the GRF proteins into two subfamilies. The various species are indicated with different colored lines. (**B**) A bar graph showing the numbers of members in the ε groups and the non-ε groups for the PbGRF, AtGRF, OsGRF, FvGRF, PavGRF, PmGRF, RoGRF, PpGRF, and MdGRF protein families.

**Figure 2 ijms-24-14690-f002:**
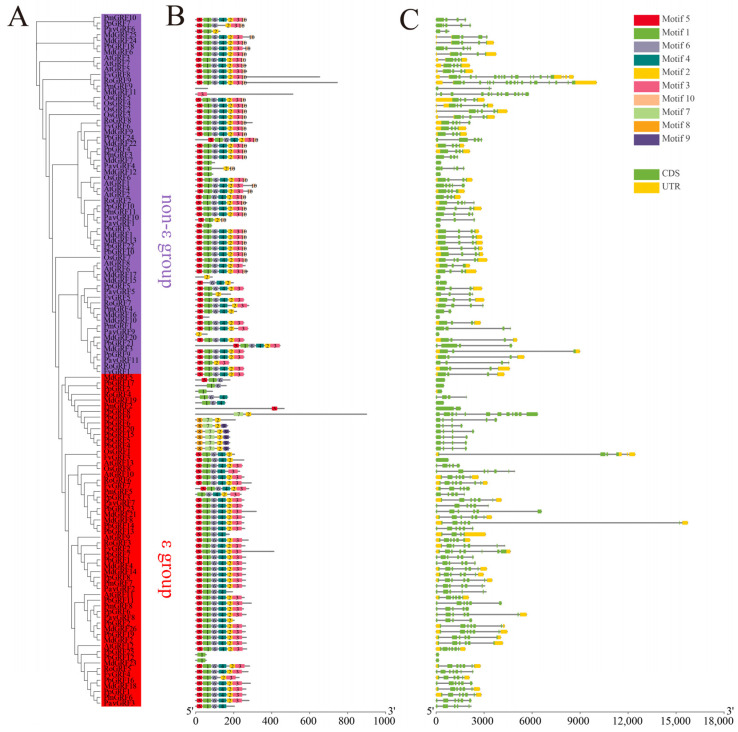
Motif distributions and gene structure of GRF genes. (**A**) Phylogenic relationships of the 121 identified GRF proteins from the nine plant species. (**B**) Motif types present in the 121 GRF proteins. The identified motifs in GRF proteins are indicated by different colored boxes and were named Motifs 1–10. (**C**) Exon–intron distribution in the *PbGRF* genes. The closed green boxes and black lines represent the exons and introns, respectively, and the closed yellow boxes represent untranslated regions (UTRs).

**Figure 3 ijms-24-14690-f003:**
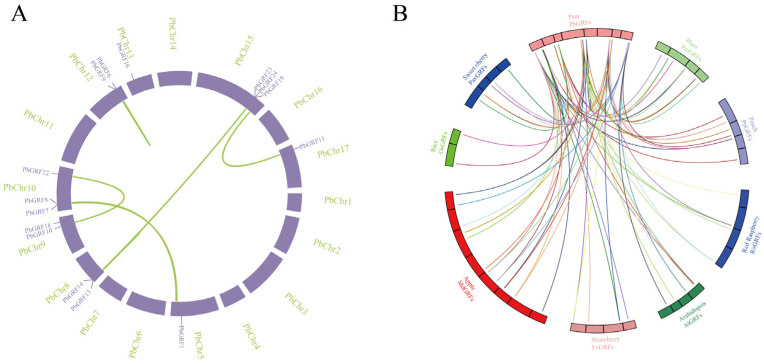
Syntenic relationships between pear and eight other species. (**A**) Chromosomal distribution and duplication relationships of PbGRF genes. (**B**) Syntenic relationships between the *GRF* genes of pear and eight other species.

**Figure 4 ijms-24-14690-f004:**
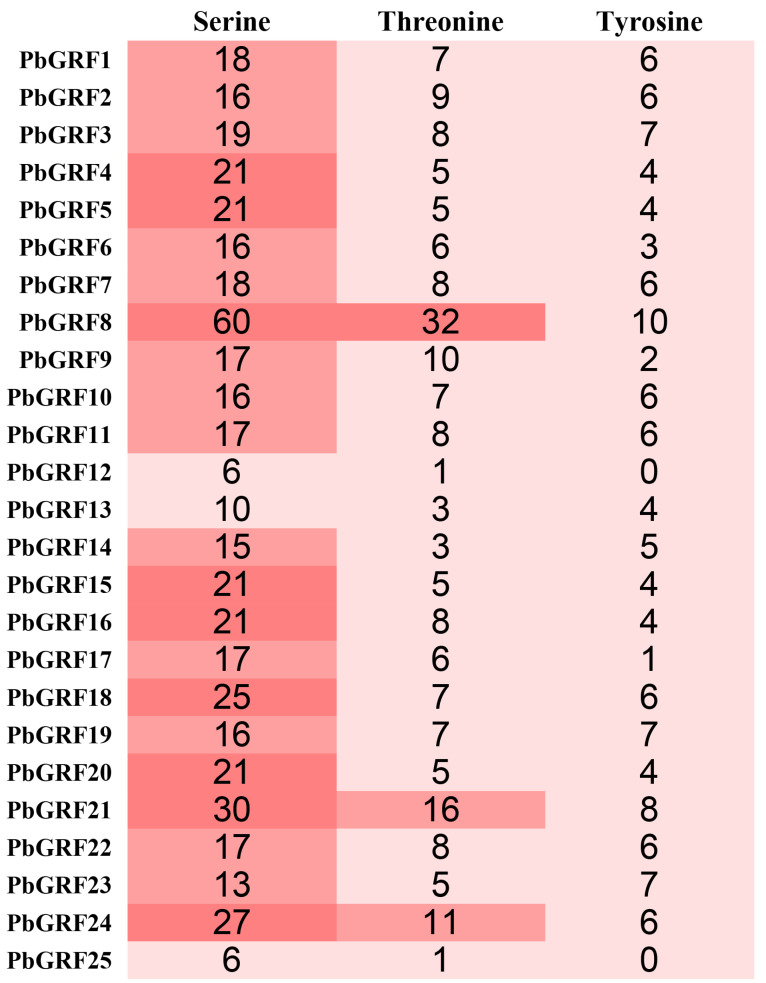
Phosphorylation site analysis of the 25 PbGRF proteins. The numbers in the first, second, and third columns, respectively, represent the numbers of serine, threonine, and tyrosine sites in the corresponding PbGRF protein sequences.

**Figure 5 ijms-24-14690-f005:**
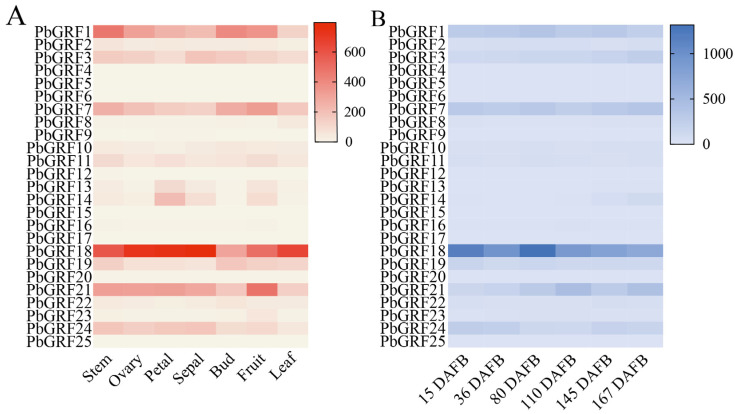
Expression patterns of the 25 *PbGRF* genes in pear tissues and fruits. (**A**) *PbGRF* gene expression in seven different tissues of pear. (**B**) *PbGRF* gene expression patterns in six developmental stages of ‘Dangshansuli’ pear fruits.

**Figure 6 ijms-24-14690-f006:**
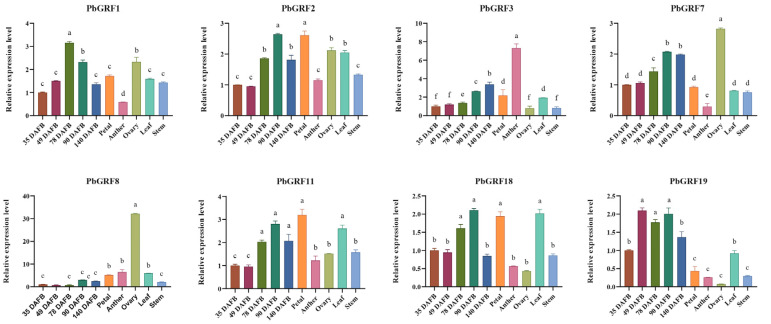
Relative expression levels of eight *PbGRF* genes in five different pear tissues and five developmental stages of pear fruits. Error bars indicate the standard errors of three technical replicates derived from a single bulked biological sample, and significant differences are determined by one-way ANOVA tests, which are indicated with small letters a–f.

**Figure 7 ijms-24-14690-f007:**
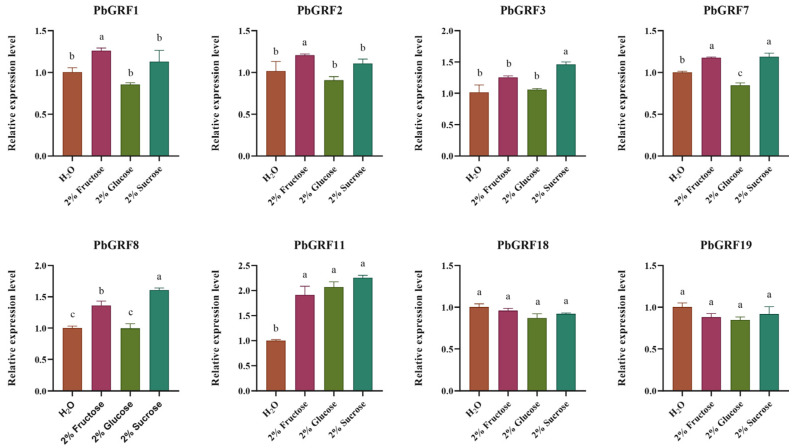
Expression of eight *PbGRF* genes in response to sugar feeding treatments in seedlings of *Pyrus betulifolia*. qRT-PCR analysis of *PbGRF* gene expression levels in the leaves in response to 2% glucose, 2% fructose, and 2% sucrose treatment. The gene expression levels shown are relative to the expression of the reference gene *PbGAPDH*. Data are presented as the mean ± standard error of three technical replicates, and significant differences are determined by one-way ANOVA tests, which are indicated with small letters a–c. Water treatment was the negative control.

**Figure 8 ijms-24-14690-f008:**
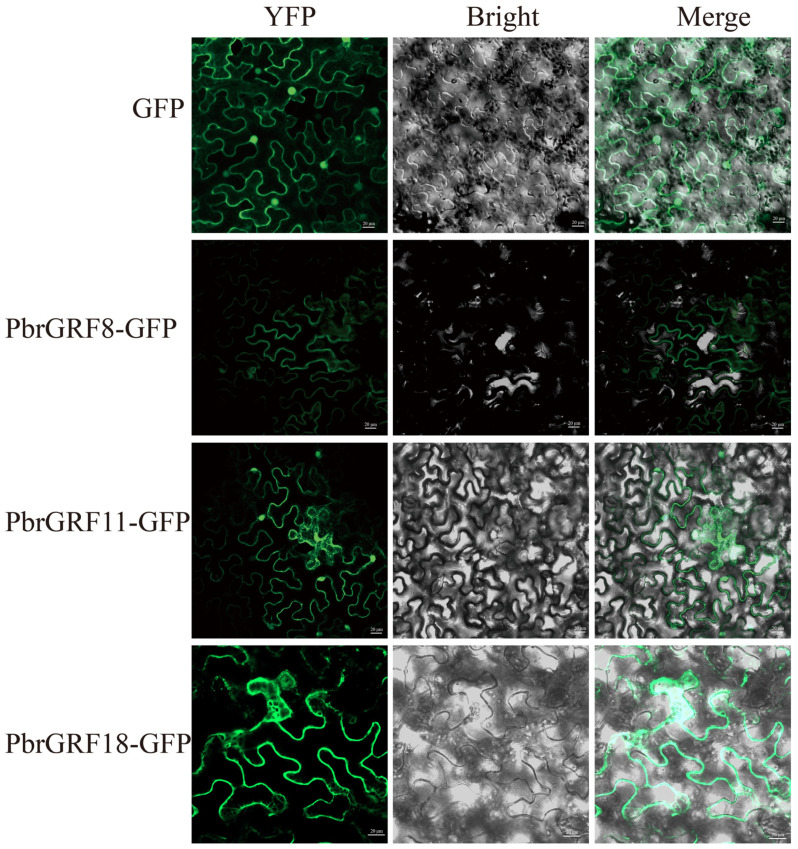
Subcellular localization of the three PbGRF proteins (PbGRF8, PbGRF11, and PbGRF18) in *Nicotiana benthamiana* leaf cells. Free GFP was used as the control. The GFP fluorescence of the three fusion proteins and the bright-field images were merged. Scale bar = 20 μm.

**Figure 9 ijms-24-14690-f009:**
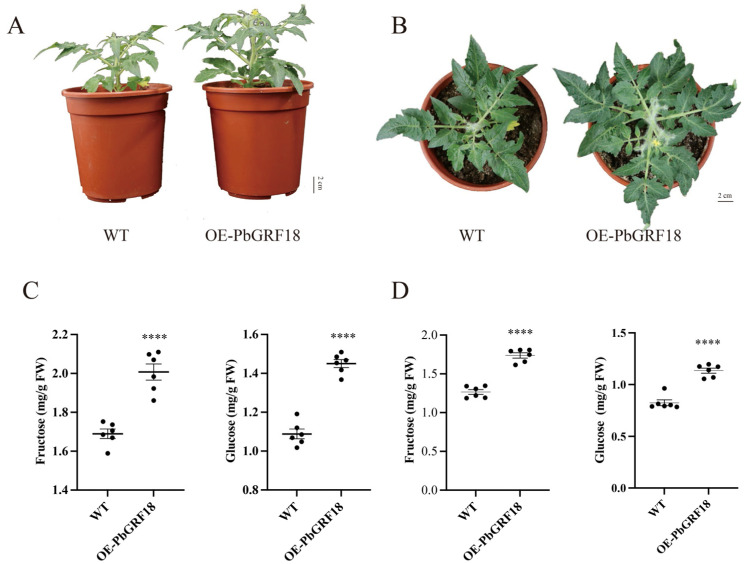
Effects of *PbGRF18* over-expression on growth of tomato cultivar ‘Micro-Tom’ and measurement of sugar contents in fruits and leaves. (**A**,**B**) Comparison of the growth of WT ‘Micro-Tom’ plants and OE-PbGRF18 plants. (**C**) Fructose (**left**) and glucose (**right**) contents of fruits of WT ‘Micro-Tom’ and OE-PbGRF18 transgenic ‘Micro-Tom’ plants. (**D**) Fructose (**left**) and glucose (**right**) contents in leaves of WT ‘Micro-Tom’ and OE-PbGRF18 transgenic ‘Micro-Tom’ plants. The significances of the differences between two groups of data were calculated using Student’s *t*-test, which were indicated with asterisks; (****) *p* < 0.0001.

**Table 1 ijms-24-14690-t001:** Information concerning GRF genes in pear.

Gene	Chr	Gene Locus	Length (aa)	MW (kDa)	pI
*PbGRF1*	Chr5	25,101,995–25,104,450	263	29.66 kDa	4.47
*PbGRF2*	scaffold1087.0	102,423–106,698	262	29.52 kDa	4.43
*PbGRF3*	scaffold1332.0	65,750–68,405	262	29.36 kDa	4.47
*PbGRF4*	Chr11	9,687,156–9,689,055	179	19.71 kDa	5.92
*PbGRF5*	Chr12	9,955,316–9,957,215	179	19.71 kDa	5.92
*PbGRF6*	Chr12	19,547,164–19,548,792	168	18.68 kDa	5.92
*PbGRF7*	Chr10	3,657,525–3,659,852	263	29.68 kDa	4.47
*PbGRF8*	Chr10	4,192,289–4,198,629	899	99.97 kDa	8.84
*PbGRF9*	Chr12	19,034,476–19,038,260	208	22.94 kDa	9.39
*PbGRF10*	Chr9	17,273,908–17,276,829	262	29.39 kDa	4.51
*PbGRF11*	Chr17	2,782,823–2,786,908	292	32.59 kDa	4.48
*PbGRF12*	Chr2	1,111,149–1,111,313	55	6.19 kDa	9.19
*PbGRF13*	Chr8	1,222,808–1,225,889	176	20.19 kDa	6.37
*PbGRF14*	Chr8	1,520,693–1,523,000	258	29.37 kDa	4.8
*PbGRF15*	scaffold489.0.1	279,520–281,468	179	19.71 kDa	5.92
*PbGRF16*	Chr13	4,575,039–4,577,295	287	32.56 kDa	4.68
*PbGRF17*	Chr2	1,234,860–1,235,339	160	18.41 kDa	8.33
*PbGRF18*	Chr9	19,676,961–19,679,128	282	31.69 kDa	4.46
*PbGRF19*	Chr15	39,603,193–39,607,232	266	29.87 kDa	4.44
*PbGRF20*	scaffold556.0	98,479–100,829	179	19.71 kDa	5.92
*PbGRF21*	Chr10	20,105,222–20,109,957	444	49.05 kDa	6.79
*PbGRF22*	Chr10	21,562,227–21,565,109	262	29.41 kDa	4.51
*PbGRF23*	Chr15	36,285,362–36,291,941	318	35.45 kDa	4.47
*PbGRF24*	Chr15	36,562,201–36,565,061	323	36.35 kDa	5.1
*PbGRF25*	scaffold956.0	67,356–67,520	55	6.19 kDa	9.19

**Table 2 ijms-24-14690-t002:** Information on GRF genes’ CREs in nine species.

Elements	PbGRF	MdGRF	FvGRF	PmGRF	PpGRF	RoGRF	PavGRF	OsGRF	AtGRF
auxin responsive element	10	20	10	6	9	9	5	2	11
cis-acting element involved in defense and stress responsiveness	12	12	2	3	4	3	2	4	12
cis-acting element involved in gibberellin responsiveness	5	4	3	0	2	0	2	0	1
cis-acting element involved in light responsiveness	6	4	1	0	2	2	1	2	4
cis-acting element involved in low-temperature responsiveness	19	24	3	7	8	9	10	2	9
cis-acting element involved in salicylic acid responsiveness	14	21	9	6	8	4	5	3	14
cis-acting element involved in the abscisic acid responsiveness	0	0	1	0	0	0	0	0	29
cis-acting regulatory element essential for the anaerobic induction	50	46	18	28	35	17	26	14	32
cis-acting regulatory element involved in light responsiveness	69	92	17	28	50	18	38	42	28
cis-acting regulatory element involved in the MeJA responsiveness	76	88	34	42	36	34	44	44	48
cis-acting regulatory element involved in zein metabolism regulation	15	16	8	5	5	7	7	8	10
cis-acting regulatory element related to meristem expression	17	11	2	5	5	8	4	5	5
gibberellin responsive element	15	20	4	5	6	9	7	5	13
light responsive element	66	30	22	15	9	11	15	11	26
MYB binding site involved in drought-inducibility	19	22	7	11	7	6	10	11	6
MYB binding site involved in light responsiveness	6	11	2	5	4	5	1	2	11
part of a conserved DNA module involved in light responsiveness	65	34	12	20	27	16	14	11	30
part of a light responsive element	70	89	31	31	24	26	39	19	37
part of an auxin responsive element	1	1	0	2	1	0	1	0	0

## Data Availability

The data and materials that support the findings of this study were available from the corresponding author upon reasonable request.

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
