# Peer review of "Genome-Wide Identification and Evolution of the GRF Gene Family and Functional Characterization of PbGRF18 in Pear"

_ijms, 2023, doi:10.3390/ijms241914690_

Round 1
Reviewer 1 Report
The authors propose a manuscript on the characterization of the GRF gene family and specifically on the PbGRF gene in the Pyrus genus. In particular, the authors report a complete analysis of the phylogeny and genetic relationships with other species of the PbGRF genes. Furthermore, they correlate the PbGRF18 gene in the transport and accumulation of sugars in the plant species studied.
The manuscript is rich in data however many critical issues remain:
1) the statistics are wrong. In fact, the data reported in figures 6 and 7 should be analyzed with one-way ANOVA and not t-student. The results should be completely reviewed.
2) The manuscript is purely descriptive. In fact, the authors describe gene families but do not answer a problem. For example, does the expression of this gene vary with abiotic or biotic stress?
3) Are there plants that do not express this gene family or the specific PbGRF18 gene?
Reviewer 2 Report
Dear Authors,
Reviewer comments ijms-2610246
The manuscript entitled „Genome-wide identification and evolution of the GRF gene family and functional characterization of PbGRF18 in pear“ represents a useful study aimed at an analysis of GRF (also called 17-3-3) gene family in pear (Pyrus betulifolia) based on genonme-wide analysis of published genomic data, sequence analysis including phylogenetic tree analysis and synteny analysis using not only pear GRFs but also GRFs from eight other plant species, cis-regulatory elements (CRE) analysis, phosphorylation site analysis, expression analysis in response to sugars treatments, and the effects of PbGRF18 overexpression in tomato „Micro-Tom“ with respect to sugars (fructose, glucose) contents in tomato leaves.
I have some minor comments on the present manuscript which are provided below:
1/ In Results, Figure 1A, bootstrap values at nodes have to be added to the phylogenetic tree scheme.
2/ In Figure 6, significant differences in the graphs have to be indicated.
3/ In Figure 7 legend, Student T-test has to be mentioned as the kind of statistical test used for evaluation of significant differences between control and sugar treatments as indicated by the asterisks.
4/ In Materials and methods, the source of the plant material used, i.e., Pyrus betulifolia seedlings, has to be given. In part 4.7., Expression profiles of PbGRF genes determined from RNA-seq datasets, appropriate references or web addresses to the softwares and online tools such as K-means clustering, pheatmap package, etc. Have to be given.
In Materials and methods, part 4.12. Statistical analyses, software used for Student T-test has to be specified.
5/ Formal comments on the text related to terminology, English language and style:
Manuscript title, line 3: Correct the typing error in the word „characterization“ (not „chararcterization“).
Abstract, line 14: Use the present form of the verb „are“ instead of „were“ in the statement: „Proteins encoded by the G-box regulating factor (GRF, also called 14-3-3) gene family are involved in protein-protein interactions…“
Abstract, line 29: Correct the term „transgenic analysis“ (not „transgenetic analysis“).
Introduction, line 36: Use the present form of the verb „is“ instead of „were“ in the statement „Pear (Pyrus ssp) is classified in the tribe Maleae….“
Line 45: Modify the statement as follows: „Additionally, sugars as a quality trait could contribute to the fruit flavor…“
Line 47: Modify the word form „relate to“ to „related to“ in the statement „Until now, only a few genes related to fruit quality were identified …“
Line 64: Add the word „which“ in the statement „Examples were the tomato GRF proteins TFT1 and TFT10 which might modulate sucrose phosphate synthase (SPS) activity…“
Results, Line 150: Modify the words „we analysis“ to „we analyse“ in the statement: „Next, we analyse the gene structure of all GRF proteins….“
Line 156: Add „the“ preceding the words „same branch“, i.e., „The results showed that GRF proteins clustered into the same branch…“
Line 275: Modify the word „regular“ to „regulate“ in the heading „2.9. PbGRF18 could regulate plant growth and fruit sugar accumulation“.
Line 284: Add the verb „has“ preceding the verb „shown“ and use „was“ instead of „with“ in the statement: „Expression analysis has shown that PbGRF18 was highly expressed in different tissues…“
Line 332: Correct the typing error in the verb „showed“ (not „howed“).
Line 338: Add the word „in“ following the verb „resulting“, correct the typing error in „membnrane“ (not „memebrane“) and modify the statement.
Line 361: Modify the statement as follows: „…indicating that the biological functions of PbGRF proteins were modified by posttranslation modification events…“
Line 363: Correct the typing error in the word „proteins“, „GRF proteins“ (not „protiens“).
Line 365: Satrt a new sentence with the words „There was a potential interaction mechanism between PbCPK28 and PbGRFs…“
Line 370: Remove „of“ in the statement „Among all PbGRFs,…“
Line 371: Correct the word order in the statement „in different pear tissues“.
Line 371: Correct the term „transgenic tomato plants“ (not „transgenetic“).
Line 376: Replace the word „and“ with „thus“ in the satetment „…might be through phosphorylated PbCPK28 thus enhancing the PbCPK28 activity…“
Final recommendation: Accept after a minor revision.

Dear Authors,
Reviewer comments ijms-2610246
The manuscript entitled „Genome-wide identification and evolution of the GRF gene family and functional characterization of PbGRF18 in pear“ represents a useful study aimed at an analysis of GRF (also called 17-3-3) gene family in pear (Pyrus betulifolia) based on genonme-wide analysis of published genomic data, sequence analysis including phylogenetic tree analysis and synteny analysis using not only pear GRFs but also GRFs from eight other plant species, cis-regulatory elements (CRE) analysis, phosphorylation site analysis, expression analysis in response to sugars treatments, and the effects of PbGRF18 overexpression in tomato „Micro-Tom“ with respect to sugars (fructose, glucose) contents in tomato leaves.
I have some minor comments on the present manuscript which are provided below:
1/ In Results, Figure 1A, bootstrap values at nodes have to be added to the phylogenetic tree scheme.
2/ In Figure 6, significant differences in the graphs have to be indicated.
3/ In Figure 7 legend, Student T-test has to be mentioned as the kind of statistical test used for evaluation of significant differences between control and sugar treatments as indicated by the asterisks.
4/ In Materials and methods, the source of the plant material used, i.e., Pyrus betulifolia seedlings, has to be given. In part 4.7., Expression profiles of PbGRF genes determined from RNA-seq datasets, appropriate references or web addresses to the softwares and online tools such as K-means clustering, pheatmap package, etc. Have to be given.
In Materials and methods, part 4.12. Statistical analyses, software used for Student T-test has to be specified.
5/ Formal comments on the text related to terminology, English language and style:
Manuscript title, line 3: Correct the typing error in the word „characterization“ (not „chararcterization“).
Abstract, line 14: Use the present form of the verb „are“ instead of „were“ in the statement: „Proteins encoded by the G-box regulating factor (GRF, also called 14-3-3) gene family are involved in protein-protein interactions…“
Abstract, line 29: Correct the term „transgenic analysis“ (not „transgenetic analysis“).
Introduction, line 36: Use the present form of the verb „is“ instead of „were“ in the statement „Pear (Pyrus ssp) is classified in the tribe Maleae….“
Line 45: Modify the statement as follows: „Additionally, sugars as a quality trait could contribute to the fruit flavor…“
Line 47: Modify the word form „relate to“ to „related to“ in the statement „Until now, only a few genes related to fruit quality were identified …“
Line 64: Add the word „which“ in the statement „Examples were the tomato GRF proteins TFT1 and TFT10 which might modulate sucrose phosphate synthase (SPS) activity…“
Results, Line 150: Modify the words „we analysis“ to „we analyse“ in the statement: „Next, we analyse the gene structure of all GRF proteins….“
Line 156: Add „the“ preceding the words „same branch“, i.e., „The results showed that GRF proteins clustered into the same branch…“
Line 275: Modify the word „regular“ to „regulate“ in the heading „2.9. PbGRF18 could regulate plant growth and fruit sugar accumulation“.
Line 284: Add the verb „has“ preceding the verb „shown“ and use „was“ instead of „with“ in the statement: „Expression analysis has shown that PbGRF18 was highly expressed in different tissues…“
Line 332: Correct the typing error in the verb „showed“ (not „howed“).
Line 338: Add the word „in“ following the verb „resulting“, correct the typing error in „membnrane“ (not „memebrane“) and modify the statement.
Line 361: Modify the statement as follows: „…indicating that the biological functions of PbGRF proteins were modified by posttranslation modification events…“
Line 363: Correct the typing error in the word „proteins“, „GRF proteins“ (not „protiens“).
Line 365: Satrt a new sentence with the words „There was a potential interaction mechanism between PbCPK28 and PbGRFs…“
Line 370: Remove „of“ in the statement „Among all PbGRFs,…“
Line 371: Correct the word order in the statement „in different pear tissues“.
Line 371: Correct the term „transgenic tomato plants“ (not „transgenetic“).
Line 376: Replace the word „and“ with „thus“ in the satetment „…might be through phosphorylated PbCPK28 thus enhancing the PbCPK28 activity…“
Final recommendation: Accept after a minor revision.
Round 2
Reviewer 1 Report
The authors responded to the critical issues raised